# Nutraceuticals in the Treatment of Pulmonary Arterial Hypertension

**DOI:** 10.3390/ijms21144827

**Published:** 2020-07-08

**Authors:** José L. Sánchez-Gloria, Horacio Osorio-Alonso, Abraham S. Arellano-Buendía, Roxana Carbó, Adrián Hernández-Díazcouder, Carlos A. Guzmán-Martín, Ivan Rubio-Gayosso, Fausto Sánchez-Muñoz

**Affiliations:** 1Sección de Estudios de Posgrado, Escuela Superior de Medicina, Instituto Politécnico Nacional, Mexico City 11340, Mexico; luis_san29@hotmail.com (J.L.S.-G.); gmcarlos93@gmail.com (C.A.G.-M.); aiorubio@gmail.com (I.R.-G.); 2Departamento de Inmunología, Instituto Nacional de Cardiología Ignacio Chávez, Mexico City 14080, Mexico; adrian.hernandez.diazc@hotmail.com; 3Departamento de Fisiopatología Cardio-Renal, Instituto Nacional de Cardiología Ignacio Chávez, Mexico City 14080, Mexico; horace_33@yahoo.com.mx (H.O.-A.); neoabraham@hotmail.com (A.S.A.-B.); 4Departamento de Biomedicina Cardiovascular, Instituto Nacional de Cardiología Ignacio Chávez, Mexico City 14080, Mexico; roxcarbo@gmail.com; 5Posgrado en Biología Experimental, Universidad Autónoma Metropolitana-Iztapalapa, Mexico City 09340, Mexico

**Keywords:** pulmonary arterial hypertension, nutraceuticals, antihypertensive, antioxidant, antiproliferative, anti-inflammatory

## Abstract

Pulmonary arterial hypertension (PAH) is a severe disease characterized by the loss and obstructive remodeling of the pulmonary arterial wall, causing a rise in pulmonary arterial pressure and pulmonary vascular resistance, which is responsible for right heart failure, functional decline, and death. Although many drugs are available for the treatment of this condition, it continues to be life-threatening, and its long-term treatment is expensive. On the other hand, many natural compounds present in food have beneficial effects on several cardiovascular conditions. Several studies have explored many of the potential beneficial effects of natural plant products on PAH. However, the mechanisms by which natural products, such as nutraceuticals, exert protective and therapeutic effects on PAH are not fully understood. In this review, we analyze the current knowledge on nutraceuticals and their potential use in the protection and treatment of PAH, as well as whether nutraceuticals could enhance the effects of drugs used in PAH through similar mechanisms.

## 1. Introduction

Pulmonary arterial hypertension (PAH) is a life-threatening disease characterized by the obstruction, loss, and remodeling of the pulmonary arteries, which increases pulmonary arterial pressure (PAP) and pulmonary vascular resistance (PVR). Hemodynamically, PAH is defined by a mean pulmonary artery pressure (mPAP) of ≥20 mm Hg, a pulmonary arterial wedge pressure (PAWP) of ≥15 mm Hg, and a pulmonary vascular resistance (PVR) of ≥three Wood Units (WU), as established at the sixth World Symposium on Pulmonary hypertension (PH) in Nice 2018 [1].

PAH prevalence has increased over time. Worldwide, the latest records show an incidence of 2.0 to 7.6 cases per million adults per year and a prevalence of 11 to 26 cases per million adults. The incidence is known to be four times higher among women, but survival is paradoxically worse in men [2,3]. PAH belongs to a group of severe clinical entities included in the current clinical classification of PH (Group 1). The other clinical scenarios include PH due to left heart disease (Group 2), PH due to lung disease or hypoxia (Group 3), PH due to chronic thromboembolic disease (Group 4), and PH due to unclear and/or multifactorial mechanisms (Group 5). In PAH, pulmonary vascular remodeling is characterized by an accumulation of vascular cells in the pulmonary arterial wall, such as changes in the pulmonary artery smooth muscle cells (PASMCs), endothelial cells, fibroblasts, myofibroblasts, and pericytes. This process can also involve the loss of precapillary arteries and perivascular infiltration of inflammatory cells (macrophages, dendritic cells, mast cells, and B and T-lymphocytes) [4]. In general, pulmonary endothelial dysfunction contributes to pulmonary vascular remodeling in all groups of PAH [5,6]. Pulmonary endothelial cells (ECs) play a key role in the regulation of pulmonary vascular tone through the nitric oxide (NO), prostacyclin (PGI_2_), endothelin (ET), and serotonin (5-HT) pathways [7]. ECs are sensitive and respond to signals from extracellular environments where interactions with adjacent cells, circulating cells, and/or mediators help maintain a thrombosis-free surface. This property helps control inflammatory cell adhesion and assure normal angiogenesis, as well as the integrity of the vascular wall [8]. Conversely, the alteration or dysfunction of the pulmonary endothelium leads to an altered balance of the vasoconstriction and vasodilation mechanisms, as well as the acquisition of proinflammatory, prothrombotic, proproliferative, and antiapoptotic phenotypes [9].

The main mechanisms for the treatment of PAH are based on an imbalance of the NO, PGI_2,_ and ET pathways [10]. Despite the existence of a wide variety of drugs for the treatment of PAH, it continues to be a life-threatening condition, and long-term treatment remains expensive. In this review, we discuss the current evidence of nutraceutical evidence in PAH models. Finally, we discuss the possibility to use combination therapy in PAH-specific drugs and nutraceuticals.

## 2. An Overview of Molecular Pathways in the Treatment of PAH

The pathogenesis of pulmonary arterial hypertension is multifactorial and complex; therefore, the relevant treatments include several interventions, such as drugs, lifestyle and dietary changes, and surgery where necessary. Current knowledge of the pathophysiology of PAH has made it possible to distinguish three main signaling pathways that play a key role in the regulation of pulmonary vascular tone (nitric oxide, endothelin, and prostacyclin). Thus, available therapies are aimed at stimulating the local production of various vasoactive substances, including nitric oxide (NO) and prostacyclin, which contribute to the regulation of vascular tone in the pulmonary arteries. The drugs used in the treatment of PAH are used directly or indirectly to preserve, stimulate, or block the signaling pathways of vasoactive substances (Figure 1).

NO, the main endogenous vasodilator, is an easily diffusible gas that is synthesized in the endothelium from L-arginine by NO synthase. It diffuses vascular smooth muscle cells and mediates vascular relaxation through stimulation of soluble guanylate cyclase, thereby generating the second messenger cyclic guanosine monophosphate (cGMP). cGMP is metabolized by phosphodiesterase 5 (PDE-5) to GMP and inhibits NO-mediated vasodilation (Figure 1). Thus, in the treatment of PAH, PDE-5 inhibitors are considered a therapeutic option directed at the NO pathway. The drugs for PAH treatment include NO—cGMP enhancers (PDE-5 inhibitors), endothelin receptor antagonists and agonists of the prostacyclin pathways, and, recently, a soluble guanylate cyclase stimulator (sGCs). Additionally, riociguat works on a distinct molecular target in the same pathway as PDE-5 inhibitors.

### 2.1. NO Pathway

In the NO pathway, NO molecule binds to soluble guanylate cyclase (sGC) and leads to the production of cyclic guanosine monophosphate (cGMP), which activates myosin light chain phosphatase (MLCPase), thereby reducing the phosphorylation of myosin to induce arteriole vasodilation and inhibit cell proliferation (Figure 1). The phosphodiesterase-5 (PDE-5) enzyme is abundantly expressed in vascular smooth muscle cells (VSMCs) of the pulmonary vasculature and upregulated in the PAH in VSMCs and right ventricle cardiomyocytes. Moreover, the upregulation of PDE-5 leads to increased hydrolysis of cGMP [11,12]. PDE-5 inhibition prevents the degradation of cGMP to GMP (Figure 1), and sildenafil and tadalafil are two typical PDE-5 inhibitors that have demonstrated clinical benefits in PAH [13].

Riociguat is a new drug that acts independently of NO via the stimulation of sGC by increasing the generation of cGMP [14,15]. sGC is expressed in VSMCs of the pulmonary vasculature, platelets, the right ventricle, and other tissues. sGC binding to NO catalyzes the conversion of GTP to sGMP, thereby promoting beneficial effects, such as inhibiting smooth muscle proliferation leucocyte recruitment, inflammation, fibrosis, platelet aggregation, and vasodilation [15].

The combined use of both drugs (PDE-5 inhibitors and the stimulation of sGC) in patients with PAH is not recommended [16].

### 2.2. ET-1 Pathway

Endothelin (ET) is a peptide or vasoactive compound and is the most potent vasoconstrictor substance known [17]. ET-1 synthesis occurs when the inactive 39-amino-acid precursor pro-ET is cleaved by ET-converting enzymes to ET-1. ET-1 has two receptor subtypes (G protein-coupled receptors): the endothelin receptor A (ET_A_R), which is located on vascular smooth muscle cells, and endothelin receptor B (ET_B_R), which has two subtypes—ET_B1_R, located on endothelial cells, and ET_B2_R, located on vascular smooth muscle cells (Figure 1). The ET_A_R expressed in VSMCs and PAH contributes to contraction, proliferation, and proinflammatory effects [18,19]. On the other hand, ET_B1_R is expressed in vascular endothelial cells, where it acts to clear ET-1 and mediate the endothelial cell vasodilation through cytosolic Ca^2+^ binding to calmodulin to activate calmodulin kinase, which is responsible for phosphorylating endothelial nitric oxide synthase (NO) and synthesizing nitric oxide NO and PGI_2_ (Figure 1) [20,21]. In vascular smooth muscle cells, ET-1 binds to ET_A_R and ET_B2_R, thereby activating phospholipase C (PLC) to release inositol trisphosphate (IP3) and diacylglycerol (DAG). IP3 binds to its intracellular receptor, thus releasing calcium (Ca^2+^) into the cytosol to produce vasoconstriction and proliferative responses [22]. In PAH, ET-1 is upregulated and promotes PASMC proliferation [23,24].

In patients with PAH, ET-1 is overexpressed in the lung and plasma [24]. Bosentan, ambrisentan, and macitentan are endothelin receptor antagonists (ERAs) that have demonstrated to have beneficial effects in PAH [25,26,27,28].

ERAs were designed as either ETA-selective antagonists (ET_A_/ET_B_ selectivity ratio >100) or dual ET_A_/ET_B_ antagonists (equal selectivity), which suggests that dual antagonists have the potential to block ET-1-induced signaling more effectively than other ERAs [22]. However, the selective antagonists block ET_A_ and vasoconstriction in VSMCs but stimulate the vasodilation mediated by the ET_B1_ receptor [29].

### 2.3. PGI_2_ Pathway

Prostacyclin (PGI2) is a lipidic mediator in endothelial cells. PGI2 is produced by prostacyclin sinthase through the degradation of arachidonic acid in endothelial cells (Figure 1). PGI_2_ is a potent pulmonary vasodilator mediator released by endothelial cells through the concerted actions of the cyclooxygenase and prostacyclin synthase. Using arachidonic acid and prostaglandin H2 as a substrate, PGI2 promotes pulmonary vasodilation and has antithrombotic and antiproliferative properties (Figure 1) [30]. The effects of PGI_2_ occur through the I-prostanoid receptor (IPR) via the activation of adenylyl cyclase, thereby converting ATP to cAMP, which increases protein kinase A (PKA) activity. In turn, PKA promotes the phosphorylation of myosin light chain kinase, which leads to smooth muscle relaxation and vasodilation (Figure 1). PKA promotes arterial relaxation by (a) phosphorylating myosin light chain kinase at a site that reduces the affinity of the kinase to the calcium–calmodulin complex, (b) activating calcium pumps in the cell membrane and sarcoplasmic reticulum, and (c) opening potassium channels through which potassium can exit and thereby hyperpolarize the cell [31].

IPRs are expressed in the VSMCs of the pulmonary vasculature [32,33]. In PAH patients, the PGI_2_ levels are reduced in the pulmonary vasculature and serum. IPRs are also reduced in patients with PAH [34,35]. Prostanoid drugs are synthetic analogs of IP that can be administered in oral, inhaled, subcutaneous, or intravenous forms. Epoprostenol, Treprostinil, and selexipag are drugs of this group that have demonstrated positive effects in the treatment of PAH [36,37,38,39,40,41,42].

Treatments for PAH are mainly focused on blocking the pathophysiological mechanisms and the activation of vasodilatory mechanisms. However, PAH is complex and multifactorial. Therefore, the ideal treatment should be focused on blocking/stimulating more than one mechanism. Indeed, in clinical practice, cotherapy and combination therapy are becoming the basis for the successful management of PAH [43]. Recently, new drugs have been introduced into clinical practice, all targeting endothelin, nitric oxide, and prostacyclin pathways, with the main objective of improving function, quality of life, and increasing survival. However, there are still a significant number of patients that do not respond adequately to or tolerate such medical therapies. Moreover, none of the available therapies have been shown to slow the progression of the disease [27,44]. On the other hand, interest in the search for the mechanisms involved and, therefore, therapeutic options continue. This review aims to provide a brief overview of other new options, whose clinical, preclinical, or basic studies have provided encouraging results, suggesting the possibility that, in the near future, they will be implemented as a therapeutic option or as a combined treatment for PAH [45].

## 3. Nutraceutical in the Treatment of Cardiovascular Disease

Nutraceuticals are frequently used for the treatment of several diseases due to their medicinal properties. One of the leading reasons underlying the search for new therapeutic alternatives is the availability and high cost of drugs for PAH. Nutraceuticals are foods or parts of food (nutrients) that provide health benefits beyond their nutritional values [46]. Some popularly consumed nutraceuticals include flavonoid plant pigments, flavonols from chocolate polyphenols (such as resveratrol from red grapes), omega-3, catechins from tea, and quercetin [47]. Nutraceuticals have antioxidant and anti-inflammatory properties that protect against severe diseases, such as cardiovascular diseases [48]. Studies have shown that nutraceuticals can be used to improve or prevent cardiovascular diseases, thereby increasing life expectancy [49,50]. However, the effects of nutraceuticals on PAH remain unclear.

The ESC/ERS Guidelines for the diagnostic and treatment of PH contain limited nutritional and lifestyle recommendations; the same is true for combination therapy with PAH-specific drugs and nutraceuticals to enhance their effects [51]. Few studies have evaluated the effects of nutraceuticals on patients with PAH. Currently, the administration of nitrate-rich beetroot juice has shown a beneficial effect in patients with PAH through an increase in pulmonary NO production [52]. Additionally, the supplementation of Vitamin C and Fe in a 40-year-old female with PAH improved her life expectancy [53]. The lack of evidence on the use of nutraceuticals in the treatment of patients with PAH is evident.

Although the incidence of PAH is higher among women, there are no nutraceutical treatment studies that compare the efficacy of such therapies between both genders. The degree of damage caused by PAH in both genders is also different; vascular remodeling and RV hypertrophy, for example, are more evident in male than female rats [1,2]. Studies suggest that these differences are related to sex hormone levels, where estrogens exert protective effects against this pathology [54,55,56]. Previous studies have demonstrated that estrogen plays an important role in regulating vascular oxidative stress in MCT-induced PAH [57]. On the other hand, in an angioproliferative model of severe PAH female rats, this pathology was associated with angioproliferative changes in the pulmonary arteries but with the absence of inflammatory and fibrinogenic processes in both the lung and heart. This was correlated with preserved RV function and may be an important factor for the better survival rates in women [58]. Based on these studies, nutraceutical treatments may offer a greater protective effect in male rats than female rats due to the different degree of damage caused by PAH between genders.

There are currently no studies investigating the consumption of cooked plants and foods to improve the symptoms caused by PAH. Despite having beneficial effects in animal models, the evidence in patients is limited. The ESC/ERS Guidelines for the diagnosis and treatment of PAH do not contain nutritional information. On the other hand, the Dietary Approaches to Stop Hypertension (DASH) is a specific dietary intervention useful for the treatment of hypertension [59]. This diet recommends the consumption of carbohydrates from green leafy vegetables (broccoli and spinach), whole grains (wheat, millets and oats), low glycemic index fruits, legumes, and beans, as well as fats (olive oil, avocados, nuts, and fish) and animal protein from lean meats, low-fat dairy, eggs, and fish. Compared to PAH, this field of therapeutic intervention is not fully studied.

The route of administration of these nutraceuticals is not discussed in this study. However, in several other studies, the administration of nutraceuticals was carried out by oral administration (through drinking water, basal diet, or tube) and injection (subcutaneous and intraperitoneal). In PAH models, no reports were found for other routes of nutraceutical administration, such as an enteral route or inhalation. In this field, there are no studies investigating the application of nutraceuticals in different routes. However, as mentioned above, nutraceuticals are commonly purchased from food and consumed via the oral route. On the other hand, routes of administration have various advantages and disadvantages. Injection ensures a controlled dose of locally administered nutraceuticals. However, in rats, there is a limited volume that can be injected. On the other hand, in patients with PAH, the oral consumption of nutraceuticals can be controlled autonomously. In PAH models, oral administration by gastric tube allows a more rigorous control of the administered dose compared to the nutraceuticals present in the basal diet [60]. However, oral administration must take into account the stability of the compound to preserve its bioactivity [61].

Some foods may alter the absorption, distribution, biotransformation, and excretion of drugs. Moreover, various interactions may have an influence on the success of drug treatment. That is to say, interactions are not always harmful to therapy and in some cases can be used to improve the drug’s effects [62]. Many medicinal plants and natural compounds have beneficial effects in the treatment of PAH [63]. Interestingly, these compounds do not have more side effects than chemical drugs [64]. In humans, the oral administration of quercetin did not show any adverse effects at doses up to 730 mg per day over one month [65]. Specifically, nutraceuticals and other natural therapies for PAH have some interest as a way to enhance the effects of typical drugs used in PAH to improve the quality of life of patients suffering from this pathology. So far, there are no reports on the interactions between nutraceuticals and the drugs used to treat PAH. No adverse effects from the consumption of these nutraceuticals have been reported either.

Despite the various investigations including a known quantity of nutraceuticals in food [66], the recommended daily doses for patients with PAH have not been studied. For this reason, there is no study that analyzes the adequate amount of fruits, vegetables, cereals, and meats that a patient with PAH should consume daily. This field of research is still unknown. Some studies have tried to determine this relationship in other pathologies, such as in the habitual intake of polyphenols and the incidence of cardiovascular events. In this case, the intake of foods with a high content of polyphenols was positively associated with a decrease in cardiovascular risk. However, more clinical trials are needed to confirm this effect to establish precise dietary recommendations for this pathology [67].

On the other hand, phytochemicals such as apple polyphenols, baicalein, berberine, magnesium, quercetin, and resveratrol have been tested in PAH animal models. These phytochemicals induce their therapeutic effects through different pathways (the antiproliferation, antioxidant, antivascular remodeling, vasodilator, apoptosis induction, and anti-inflammatory pathways), which are similar to those of PAH-specific drugs [68,69,70,71,72,73]. Although several studies have shown that nutraceuticals have significant positive effects on PAH, there is a lack of evidence of these effects in patients.

## 4. Nutraceuticals in the Treatment of Pulmonary Arterial Hypertension

### 4.1. Genistein

Genistein (5,7-dihyroxy-3-(-4-hydroxyphenyl)chromen-4-one) is an isoflavone abundant in soybean and soybean products. Genistein was first extracted from a dyer’s broom (*Genista tinctoria*) in 1899. Several foods have variable amounts of this isoflavone. It can be found in different legumes, such as chickpeas (garbanzo beans), and in soy-based newborn child formula, soy milk, soy flour, soy protein isolate, alfalfa, clover sprouts, barley meal, broccoli, cauliflower, sunflower seeds, and clover seeds [74]. Based on intake from diet, legumes are the second most important source of genistein at 0.2 to 0.6 mg/100 g [75]. Genistein exerts beneficial effects on the vasculature, presumably due to its hypolipidemic, weak estrogenic, and antioxidative effects [76].

Several cooking methods affect the genistein integrity in food, such as high cooking temperatures and the fermentation used during soy food production, which can exacerbate the loss of isoflavones in soy-based products compared to the oven drying and explosive puffing processes [77,78]. Another study reported that genistein contents were reduced at 100 °C for 20 to 60 min [79]. Additionally, enzymatic hydrolysis lowers the content of isoflavones in soy-based foods, in contrast with boiling and grinding, which do not affect the amount of genistein [80]. The amount does not change if there is no mass loss during soy food production. Thus, certain processing conditions and methods, such as a low cooking temperature over an optimum cooking period, might be suitable for the retention of isoflavone in soy-based products [81].

After oral administration, genistein is absorbed after deglycosylation is catalyzed by the colon microflora or the enzyme lactase phlorizin hydrolase in the small intestine [82]. The absorbed genistein is then conjugated, forming glucuronide and sulfate conjugates in the plasma. The genistein peaks in the plasma are reached after 5–8 h [83]. Genistein is excreted in the urine and feces in the form of metabolites such as dihydrogenistein, 60’eOHeO-desmethylangolensin, trihydroxybenzene, and 30,40,5,7-tetrahydroxyisoflavone [84]. There is evidence that 50 mg per day genistein administered orally offers the best clinical efficacy [85]. Optimum steady-state serum isoflavone concentrations consumed throughout the day are more effective than those from a single highly enriched product [86]. To find another way to supply genistein, Rassu et al. (2008) used an intranasal method to deliver genistein. Genistein-loaded chitosan nanoparticles were successfully obtained and were able to promote the passage of genistein through the nasal mucosa. These results are promising as a way to supply genistein to the blood flow, but further investigations need to be performed [87]. Genistein exerts protective vasodilation, cardioprotective, and anti-inflammatory effects that are mediated by its high affinity for estrogen receptor b (ERβ) (Figure 2) [88,89]. The structure of genistein is similar to that of estradiol (i.e., its phenolic ring base and the distance between the 4’- and 7’-hydroxyl groups). Genistein also has the ability to bind to sex hormone-binding proteins and estrogen receptors [90]. In this context, genistein exerts effects on breast and prostate cancer through direct competition with estrogen to bind to the estrogen receptor; prolonged exposure decreases these receptors’ expression, thus lowering endogenous estrogen responsiveness [91]. Genistein also acts as an antioxidant via an increase in NO production through the ERK1/2, PI3 Kinase/Akt, and AMP/protein kinase A signaling pathways, which results in the phosphorylation and activation of eNOS [92,93].

In the field of PAH, two recent studies using MCT (50 mg/kg and 60 mg/kg) showed that genistein improves RV function, pulmonary vascular resistance, and survival in a rat model. The first study proposed that genistein’s action is mediated by estrogen-β receptors (ERβ), due to genistein’s ability to restore the expression of the estrogen-β receptor in the RV and the lung [94]. In this study, after 28 days of MCT injection, rats received a daily subcutaneous dose of 1 mg/kg genistein for 10 days. The effect of genistein is also related to RV hypertrophy, the proliferation of PASMCs, and the endothelial cells in the lung and heart. On the other hand, the second study showed that genistein induces its effects by modulating the PI3K/Akt/eNOS signal pathway [95]. After PH was established by MCT injection (after 28 days), the rats received subcutaneous doses of 20, 80, and 200 μg/kg genistein every day for 14 days. Additionally, genistein attenuates or prevents the effects of low temperatures in the induction of PH and ascitic syndrome in broiler chickens [96,97]; genistein was reported to be associated with this syndrome through an endothelial function by modifying two vasoactive molecules: endothelial NO and ET-1. In another study, genistein attenuated the hypoxia-induced hypertrophy of PASMCs [97].

The evidence that genistein improves the effects of PAH on MCT-induced PAH in an animal model suggests that dietary management with genistein rich legumes or soy supplements may be used to complement or enhance the pharmacological treatment of these patients. However, more clinical studies that explore the bioavailability and efficacy of genistein are necessary to achieve health benefits.

### 4.2. L-arginine

L-arginine (2-Amino-5-guanidinopentanoic acid) is a proteinogenic amino acid found in 3% to 15% of all food proteins. Proteins from soy, peanut, walnuts, and fish are rich in L-arginine. L-arginine has biomedical relevance as a precursor for NO synthesis. Arginine is the most strongly affected amino acid by cooking methods [98]. After oral administration, L-arginine is highly metabolized in the small intestine and the liver by arginase [99]. The above is involved in a very short half-life of about 1 h [100]. A previous study reported that approximately 1% of L-Arginine administration was utilized as a substrate of NOS [101]. In plasma, L-arginine is easily taken up through amino acid transport system y+ by the endothelial cells [102], which is colocalized with eNOS in the caveolae [103]. L-arginine interacts with NOS and is oxidized to form NO and L-citrulline [104]. The biological effects of NO are produced through an increase in intracellular cGMP, which is responsible for several effects of NO, including vasodilation [105].

NO is a crucial molecule in maintaining vascular homeostasis due to its potent ability to relax the arteries, prevent coagulation, aggregate platelets, and antagonize the proliferation of vascular smooth muscle cells [106]. In the cardiovascular system, NO is generated by endothelial nitric oxide synthase (eNOS) in the coronary microvessels and has protective effects directly on the endothelium and the vascular muscle, thereby promoting cardioprotection [107]. To the best of our knowledge, studies that compare L-arginine supplementation between genders are rare. Another study reported that a single intravenous dose of 30 mg of L-arginine induced vasodilation in humans [108]. Two studies on primary pulmonary hypertension patients did not observe this effect [109,110]. A recent study reported that L-arginine supplementation (6000 mg/day during 12 weeks) with a simple walking regimen is safe and efficacious for clinically stable PAH patients and yields an improvement in VO_2_max, time-to-VO_2_max, VO_2_ at the anaerobic threshold, HR recovery, and SF-36 subscales of Physical Functioning and Energy/Fatigue [111]. In rats, L-arginine treatment (500 mg/Kg) decreased RVSP, right heart hypertrophy (Figure 2), mortality, superoxide anion $O_{2}^{-}$ generation, and pulmonary artery wall thickness and increased NO production in MCT-induced PAH (Figure 3). The same studies found that L-arginine preserved eNOS expression/phosphorylation and maintained the association between eNOS and HSP90, which facilitated the restoration of eNOS activity and coupling activity to maintain the balance between NO and $O_{2}^{-}$ [112]. In another study on rats that underwent a left-sided unilateral pneumonectomy and the simultaneous implantation of a telemetry catheter into the common pulmonary artery trunk, seven days after surgery, PAH was induced by a single subcutaneos injection of 60 mg/kg MCT. The combined administration included 300 mg/kg of L-arginine and 20 mg/kg of tetrahydrobiopterin through oral gavage. L-arginine and tetrahydrobiopterin are substrates in the production of NO [113]. After 28 days of MCT-induced PAH, therapy was administered daily over 14 consecutive days. Finally, L-arginine, tetrahydrobiopterin, or both equally, decreased mPAP, attenuated RV hypertrophy, increased pulmonary vascular elasticity, and prevented body weight loss. Interestingly, the combined administration of both substances did not reveal any synergistic therapeutic effects.

Future studies should focus on selecting patients carefully for supplementation with L-arginine or increase the intake of foods rich in this amino acid. This would facilitate testing the quality of life improvements among these patients outside of pharmacological treatments.

### 4.3. Berberine

Berberine (BBR) is a quaternary ammonium salt that is considered an alkaloid (5,6-dihydrodibenzoquinolizinium derivative) and is extracted from several medicinal herbs, such as *Cortex phellodendri* and *Rhizoma coptidis*; it is found in the bark, rhizomes, roots, and stems of Berberis vulgaris L. (*Berberidaceae*). It exerts different types of biological activities, such as antioxidant, anti-inflammatory, cholesterol-lowering, antihyperglycemic, and antidepressive effects [114].

Adenosine monophosphate-activated protein kinase AMPK is an upstream enzyme of eNOS that promotes phosphorylation at the Ser1177 site, as well as NO production [115]. AMPK, as an intracellular energy receptor, has become a new target for the treatment of cardiovascular complications due to its effects on endothelial cell function. It has been reported that AMPK plays a regulating role in NO synthesis in ECs [115,116].

Zhang et al. observed that BBR ameliorates fatty acid-induced endothelial dysfunction, upregulating eNOS and downregulating NOX4 through AMPK activation [117]. In blood vessels and cultured ECs, BBR enhanced eNOS phosphorylation, attenuated high glucose-induced ROS generation, cellular apoptosis, and NF-κB activation through AMPK activation [118].

A recent study showed that BBR attenuated monocrotaline-induced pulmonary hypertension by suppressing the endothelin pathway [119].

In a very recent study using BBR in hypoxic human PASMCs (HPASMCs) and a sugen/hypoxia PAH model (Su/Hox), Wande et al. discovered that thioredoxin (Trx1) and β-catenin are overexpressed in HPASMCs during hypoxia and that BBR treatment could reduce them as efficiently as a Trx1 inhibitor (PX12), as well as silencing Txr (siTrx1) [120]. The authors also suggested that these two molecules are related and that β-catenin may be a downstream molecule of Trx1. The physiological results of the Su/Hox model showed that BBR treatment reverted all the altered parameters recorded by an echocardiogram, including mPAP, the right ventricle anterior wall, the right ventricle’s internal dimensions, right ventricle hypertrophy, and the pulmonary arterial velocity time integral. Its effects were reliable and comparable to those of PX12. The percentage of muscularization was significatively reduced, as the medial thickness did not suggest a reduction in arterial remodeling. These results thus promote Trx1 as a novel therapeutic target for PAH.

Lou et al. used a BBR dose of 100 mg/kg to prevent the proliferation and migration of PASMCs. The authors found that BBR reduces the medial wall thickness and lowers RV/(lV + S) [70].

The present studies provide evidence that BBR may be effective for PAH treatment. However, BBR’s solubility and dissolution is poor. Moreover, a high dose can cause side effects, and the treatment of PAH using BBR remains to be studied in clinical settings.

### 4.4. Naringenin

Naringenin (5,7-Dihydroxy-2-(4-hydroxyphenyl)chroman-4-one) is an aglycone of naringin and an abundant flavone commonly found in citrus fruits, such as grapefruits, oranges, and tomatoes [121]. After oral administration, naringenin is hydrolyzed through β-glucosidase in the small intestine [122], and its absorption occurs through two transport mechanisms: passive diffusion and active transport [123]. However, naringenin is poorly absorbed in the gastrointestinal tract [124]. In plasma, naringenin is found in two conjugates, the glucuronides and sulfates, and is circulated to the liver, cerebrum, kidney, spleen, and heart [125]. Subsequently, naringenin is mainly metabolized to naringenin-o-β-D-glucuronide in the liver, which comprises 98% of this metabolite detected in plasma [126]. Naringenin excretion occurs through both the biliary and urinary pathways. A study reported that naringenin metabolites like naringenin 7,4′-disulfate, and naringenin 4′-glucuronide are excreted in urine, whereas naringenin 7-glucuronide 4′-sulfate, and naringenin 7-glucuronide are excreted by bile [127]. The antioxidant effects of naringenin occur through the inhibition of xanthine oxidase, nicotinamide adenine dinucleotide phosphate oxidase, lipoxygenase and cyclooxygenase, metal ion chelation, and the scavenging of free radicals. Moreover, naringenin increases the activity of glutathione peroxidase, superoxide dismutase, and catalase. It is also known to reduce the protein nitration and oxidation facilitated by peroxynitrite [128]. The anti-inflammatory effects of naringenin occur through the activation of Nrf2, which is involved in decreasing the formation of reactive oxygen species and inflammatory mediators [129]. The cardioprotective effects of naringenin occur through the modulation of NO and oxidative stress [130]. Further, naringenin blocks human ether-a-go-go-related gene potassium channels [131].

After consuming cooked tomato paste containing 3.81 mg naringenin, the peak plasma concentration of naringenin was 0.12 ± 0.03 µmol/L 2 h [132]. After the oral consumption of orange and grapefruit juice, the peaks of naringenin in plasma were found between 4 and 6 h [133]. Naringenin has poor water solubility, and pH affects its stability. Previous studies reported that the solubility of naringenin in citrus increases at temperatures between 20 and 90 °C and at pH 3.5–8.5 [134]. Whether the citrus fruit is consumed raw, cooked, stewed, or boiled determines the beneficial effects of naringenin through the related thermal processes. However, heat-treated naringenin enhances cellular antioxidant activity, stimulates the cytotoxic activity of natural killer cells, and also decreases the cytotoxicity of T cells [135].

Naringenin has antiproliferative, anti-inflammatory, antihypertensive, and antioxidant properties. Naringenin has also shown antihypertensive, vasorelaxant effects, and reduced lung metastasis in a breast cancer resection model [136,137]. To the best of our knowledge, studies that compare naringenin supplementation between genders are lacking.

Naringenin in combination with L-arginine (a single subcutaneous injection of MCT (60 mg/kg), followed by 500 mg/kg of L-arginine and 50 mg/kg of naringenin, was orally administered daily for three weeks, thereby inducing a protective effect on the hemodynamic parameters. Naringerin also reduced the eNOS, apoptotic markers, and inflammation (Figure 3). The addition of naringenin may improve or enhance the protective effects of L-arginine in MCT-induced PAH in rats. Thus, it is necessary to also study the protective effects of supplementary naringenin in PAH patients [138].

### 4.5. Ellagic Acid

Ellagic acid (2,3,7,8-Tetrahydroxy-chromeno[5,4,3-cde]chromene-5,10-dione) is a derivative of gallic acid generated by the hydrolysis of ellagitannins [139]. Ellagic acid is found in strawberry, raspberry, blackberry grapes, green tea, and pomegranates [140]. Chemically, ellagic acid is a highly thermostable molecule due to its four rings that represent the lipophilic domain and four phenolic and two lactone groups that represent the hydrophilic zone [141].

Studies on raspberry seeds showed that ellagic acid is stable up to 200 °C [142]. Furthermore, it was reported that storage at high temperatures along with fermentation decreased the ellagic acid concentration in wine. In contrast, grape juice had a higher concentration of this polyphenol [143]. Studies indicate that only a limited fraction of ellagic acid is orally bioavailable [144] because ellagic acid has low aqueous solubility, is metabolized in the gastrointestinal tract, undergoes irreversible binding to cellular DNA and proteins, and experiences a first-pass effect [145]. Once ellagic acid is ingested, it peaks after 1 h and is eliminated after 4 h [146]. Dietary pomegranate juice and extracts exhibit antioxidative activities [147]. Ellagic acid is a strong antioxidant that attenuates the damaging effects of H_2_O_2_ by scavenging superoxide anions and hydroxyl anions [148]. Some experiments have shown that 20 μg/mL of ellagic acid inhibits radical generation of the superoxide anion $O_{2}^{-}$ [149]. Ellagic acid is a naturally selective estrogen receptor-α and estrogen receptor-β ligand, exhibiting selective estrogen receptor modulator properties similar to those of synthetic natural selective estrogen receptor modulators [150]. However, to the best of our knowledge, studies that compare ellagic acid supplementation between genders are lacking.

In rats with PAH induced by a single intraperitoneal dose of MCT (60 mg/kg), two ellagic acid dosages of 30 and 50 mg/kg daily for four weeks after MCT reduced the hemodynamic parameters (RVSP), RV hypertrophy (Figure 2), and wall thickness of the pulmonary arteries. Ellagic acid also inhibited the oxidative stress from NLRP3 inflammasome activation, decreased caspase-1 and IL-1β in the lung, and reduced inflammatory cytokines in the serum [151].

Ellagic acid needs more studies on its effects compared to those of pharmacology therapies used in the treatment of PAH, as well as the exact doses needed to induce protective effects against PAH.

### 4.6. L-citrulline

L-citrulline (2-Amino-5-(carbamoylamino)pentanoic acid) is a physiological nonessential alpha-amino acid that is an important component of the urea cycle in the liver and kidney [152]. L-citrulline is found in high concentrations in watermelon (1.6 to 3.5 mg/kg in a fresh watermelon) [153]. After oral administration of L-citrulline, a peak in plasma was detected after 1 h and returned to baseline within 5–8 h [154]. L-citrulline transport in the intestine is poorly understood; however, some studies have suggested that L-citrulline transport is involved in a Na-dependent system [155]. Additionally, the transport system of L-citrulline seems to involve B^0,+^ (Na^+^-dependent), L system (Na^+^-independent), and b^0+^ (Na^+^-independent), with comparable activity [156]. L-citrulline exerts protective effects against endothelial damage by upregulating eNOS expression, which improves endothelial function and plays an atheroprotective role [157]. L-citrulline also increases high-density lipoprotein (HDL) levels, reduces serum AST/ALT, and promotes structural changes in the endothelial structure of the thoracic aorta [158]. Further, L-citrulline exerts antioxidant activity through a reduction in hydroxyl radical formation independent of NO by directly interacting with hydroxyl radicals via the alpha-amino acids in its protonated NH3 state leading to water formation [159].

A previous study found that oral administration of L-citrulline increases L-arginine plasma concentrations [154]. L-citrulline is taken by the kidney and metabolized into L-arginine [160]. L-citrulline may serve as an L-arginine precursor as argininosuccinate synthase enzyme converts L-citrulline to L-argininosuccinate and subsequently to L-arginine via the argininosuccinate lyase enzyme [152]. Watermelon is naturally rich in L-citrulline, whose metabolic pathways, synthesis, and catabolism are regulated in the vegetative tissues of this fruit during drought stress. Different environmental conditions do not affect its stability due to the significance of its metabolism during abiotic stresses, such as drought and nitrogen limitations, in watermelon [161].

L-citrulline has several positive therapeutic effects like the potential impact on protection against endothelial damage, antioxidant and anti-inflammatory effects, effects on blood flow and blood pressure, and arterial stiffness [162]. Furthermore, it was found that the minimal effective dose of L-citrulline is 3 g/kg and the maximum effective dose is 10 g/kg [163]. To the best of our knowledge, studies that compare L-arginine supplementation between genders are lacking.

To study PAH, hypoxic piglets received oral L-citrulline starting on day three of hypoxia and continuing throughout the remaining seven days of hypoxic exposure. For the lowest dose, 0.13–0.26 g/kg L-citrulline was administered orally with a syringe twice daily. For the highest dose, 0.26 g/kg of L-citrulline was administered, and an additional L-citrulline (0.5–1.0 g/kg) was also mixed into the milk, which was consumed throughout the day. Pulmonary vascular resistance was lower in the L-citrulline-treated hypoxic piglets. This effect could be due to NO production and eNOS in the pulmonary arteries of the L-citrulline-treated piglets (Figure 3) [164].

Oral supplementation and L-citrulline-rich diets may be another strategy to protect PAH. Additionally, the administration of L-arginine and L-citrulline may represent an adjuvant strategy to improve the pharmacology effect of drugs in patients with PAH. However, the mechanisms by which it provides its beneficial effects as a nutraceutical are currently unknown.

### 4.7. Capsaicin

Capsaicin ((6E)-N-[(4-Hydroxy-3-methoxyphenyl)methyl]-8-methylnon-6-enamide) is a crystalline, lipophilic, colorless, and odorless alkaloid. Capsaicin is the active ingredient in chili peppers and gives them their characteristic pungent flavor. Capsaicinoids are responsible for the spicy flavor of the chili pepper fruit, whose main capsaicinoid is capsaicin [165]. For red pepper, cooking methods like stir-frying and roasting are preferred to retain the nutrient composition and antioxidant properties of the fruit [166]. In contrast, boiling and steaming are not recommended for this food. After intake, capsaicin is quickly absorbed in the stomach and the small intestine and is then metabolized in the liver in around 20 minutes. However, many different organs, such as the lung and skin, can also metabolize this compound [167,168,169,170].

Research has shown that capsaicin is a powerful antioxidant; it can attenuate the inflammatory response and possesses gastrointestinal and cardiovascular protective properties [171,172]. The mechanism of action of capsaicin has been extensively studied. Capsaicin can bond to the transient receptor potential vanilloid 1 (TRPV1), which is a nonselective cation channel that is mainly expressed in the sensory neurons [173]. However, studies revealed that TRPV1-positive afferent fibers are present throughout the respiratory tract [174]. Evidence suggests that capsaicin consumption induces similar effects between genders.

MCT-induced PAH rats received three subcutaneous injections of capsaicin (50, 100, and 150 mg/kg) daily to alleviate pulmonary inflammation. In the first two days, capsaicin administration was performed eight times per day in small doses with at least a one-hour interval, and on the third day, capsaicin was administered four times with a two-hour interval. Subsequently, the rats were administered MCT (60 mg/kg intraperitoneally). To determine whether capsaicin pretreatment provided beneficial effects through the p38mitogen-activated protein kinase (p38MAPK) pathway. The p38MAPK inhibitor SB203580 was administered intraperitoneally at 20 mg/kg/d after MCT injection, and the p38MAPK activator P79350 was administered at 1 mg/kg/d. All processes were carried out for 28 days. Finally, the capsaicin reversed RVSP and RV hypertrophy and alleviated inflammation (Figure 1). The upregulation of p38MAPK was reversed via capsaicin pretreatment, and the inhibition of p38MAPK provided the same benefits as capsaicin pretreatment (Figure 3). This study demonstrated that capsaicin pretreatment reversed PAH by alleviating inflammation via the p38MAPK pathway [175].

### 4.8. Xanthohumol

Xanthohumol ((E)-1-[2,4-Dihydroxy-6-methoxy-3-(3-methylbut-2-enyl)phenyl]-3-(4-hydroxyphenyl)prop-2-en-1-one) is the most abundant flavonoid in Hops (*Humulus lupulus* L.) flowers, with content of 0.1–1% (dry weight) [176]. Beer is a dietary source of xanthohumol; however, most commercial beers have less than 0.2 mg/L, which is not enough to be beneficial [177]. An efficient method to isolate and purify xanthohumol from hops extracts is via the high-speed counter-current chromatography method [178]. The enrichment of xanthohumol in beer is complicated because this flavonoid is poorly soluble in water and undergoes isomerization during wort boiling [179]. Dark beers contain higher quantities of xanthohumol because of their roasted ingredients. Further, the late addition of xanthohumol and rapid cooling of the wort reduces the losses of xanthohumol by isomerization. Xanthohumol can be biotransformed to glucuronides, hydroxylated metabolites, and cyclic dehydro-metabolites. When xanthohumol is fed to rats at a dose of 1000 mg/kg, feces is the main route of excretion [176]. The bioavailability of xanthohumol is dose-dependent and has a distinctive biphasic absorption pattern, featuring slow absorption after oral administration in humans. Enterohepatic recirculation, moreover, contributes to the long half-life of xanthohumol [180]. Xanthohumol is related to hypoglycemic, antihyperlipidemia, cancer chemo-preventive, antiangiogenic, proapoptotic, antimicrobial, and antiparasitic activities [181]. Xanthohumol has received much attention due to its anti-inflammatory and antiproliferative properties exerted by interfering with the VEGF and MAPK signaling pathways [182]. There is no evidence that suggests any differences in xanthohumol supplementation between genders.

A recent study examined the effects of xanthohumol-fortified beer on MCT-induced PAH [183]. The beer used in this study was a Portuguese beer named “Superbock” fortified with xanthohumol up to a final concentration of 10 mg/L. Experimental PAH was induced with a single subcutaneous injection of MCT (60 mg/kg). Then, after injection, rats had ad libitum access to xanthohumol-fortified beer until the last day of experiments (28 days). Pulmonary and cardiac improvements were observed with an increase in the physical capacity and survival of the rats (Figure 2) through the modulation of the ERK1/2 and PI3K/AKT pathways in the vasculature (Figure 3). Xanthohumol-fortified beer also improved RV function and remodeling through a decrease in VEGFR-2 expression.

Because beer is one of the most popular alcoholic beverages in the world, these fortification strategies could have several health benefits. However, the improvements observed in rats must be examined carefully when translating the results to human disease (to determine the beneficial dose and if humans can metabolize the substances similarly). In the future, it will be important to determine if fortified beer intake is beneficial for PAH.

### 4.9. Chrysin

Chrysin (5,7-Dihydroxy-2-phenyl-4H-chromen-4-one) is a flavonoid belonging to a flavone class of polyphenolic compounds with 15-carbon skeletons. The major natural sources of this flavonoid include passion fruit, honey, and propolis [184]. In the forest, honey may contain up to 5.3 mg/kg chrysin [185]. Currently, there are no reports on factors such as temperature or cooking methods that affect the content of chrysin in propolis (28 g/l), *Passiflora edulis* (0.012–0.120 mg/ml), molasses (5.3 mg/kg), or forest honey (0.10 mg/kg) [186]. Chrysin, like other polyphenols, experiences poor absorption and fast elimination. First, chrysin is hydrolyzed by intestinal enzymes or colonic microflora, which degrade the aglycones and release various aromatic acids. During absorption, naturally occurring chrysin becomes conjugated (via methylation, sulfation, and glucuronidation) during its passage across the small intestine and then in the liver as Phase-II conjugation gives rise to glucuronides, sulfates, methyl conjugates, and small quantities of free aglycones [187]. The biological activities of chrysin are associated with anti-inflammatory and antioxidant effects [184]. There is no conclusive evidence that chrysin consumption has differences between genders.

In cardiovascular health, chrysin ameliorates myocardial damage in rats with a dose of 25–50 mg/kg [188]. To evaluate the efficacy of chrysin in PAH, 5 mg/kg of alpha-naphthylthiourea (ANTU; a rodenticide that induces lung toxicity) was administered intraperitoneally daily for four weeks to induce PAH. Subsequently, the rats were treated with chrysin (10, 20, and 40 mg/kg orally) for four weeks. In this study, chrysin attenuated eNOS and upregulated the VEGF RNAm and protein in the RV and pulmonary artery. Thus, the protective effects of chrysin on PAH may occur through the modulation of inflammatory responses (5-HT, LDH, and GGT), oxidative stress and VEGF, and eNOS levels (Figure 3). There are other studies on the dietary implementation of chrysin as a preventive measure for cardiovascular diseases. This compound can thus be included in the diets of PAH patients.

### 4.10. Blueberry Extract

The blueberry (genus *Vaccinium L*.) is a native American species, which has high quantities of anthocyanins and polyphenolic compounds [189]. Anthocyanins are flavonoids and glycosides composed of anthocyanidin aglycone and a sugar moiety. Blueberries are an excellent source of anthocyanins, and currently it is understood that blueberries have the greatest antioxidant abilities among fruits and vegetables [190]. Blueberries that are baked, microwaved, simmered, pan-fried, or frozen present no differences in their antioxidant capacities after cooking. Pan-fried blueberries have a significantly greater antioxidant capacity than baked or simmered blueberries. Therefore, the antioxidants in blueberries appear to be heat stable. Cooked wild blueberries are thus recommended as a good source of dietary antioxidants compared to uncooked fruit [191]. Some studies indicate that anthocyanins are rapidly absorbed in the stomach and small intestine of rats and are found in the blood circulation and urine in intact, methylated, glucuronide, and/or sulfate conjugated forms. The absorption mechanism for anthocyanins is mediated by the organic anion membrane carrier bilitranslocase located in the stomach and small intestine [192]. Blueberries are great sources of anthocyanins and have antioxidant activity. Several studies mention that anthocyanins inhibit the expression of genes linked to proinflammatory proteins. Moreover, blueberry intake improves endothelial functions in subjects with metabolic syndrome and reduces blood pressure [193,194]. To the best of our knowledge, studies that compare blueberry extract administration between genders are lacking.

Recently, blueberry extract was used in an experimental model of PAH induced by MCT (60 mg/kg, intraperitoneally) to evaluate its effects on functional parameters and antioxidant stress [195]. Rats were treated with blueberry extract at doses of 50, 100, and 200 mg/kg via gavage for five weeks (two weeks before MCT and three weeks after MCT injection). This study showed that 100 mg/kg of blueberry extract improved the diastolic and systolic RV function and also decreased the mPAP (Figure 1). The blueberry extract was also able to attenuate the oxidative alterations caused by PAH. Moreover, the blueberry extract restored the expression of the antioxidant transcriptional factor Nrf2 and the endothelin receptor (ETA/ETB) (Figure 3).

Therefore, antioxidant-rich foods like blueberries in a regular diet act as a protective or preventive alternative to improve the quality of life of PAH patients. This antioxidant extract may be used as a nutraceutical to enhance the effects of standard therapy for PAH.

### 4.11. Quercetin

Quercetin (2-(3,4-dihydroxyphenyl)-3,5,7-trihydroxy-4*H*-chromen-4-one) is one of the most abundant naturally occurring polyphenols in food. Naturally, quercetin is present primarily as glycoside in many fruits, such as apples, cranberries, cherries, and grapes, and in vegetables, such as as onions, peppers, and asparagus. It is also present in other foods such as wine and black or green tea. Onions are the most important source of quercetin in the human diet [196]. In Western diets, quercetin intake has been observed in the range of 3–40 mg, while high-intake quercetin consumers consume 250 mg/day [197]. Boiling and onion lose about 30% quercetin, while microwave cooking without water better retains the flavonoids [198]. Moreover, about 50% of quercetin is readily transferred from onion into soup during cooking. Because the onion’s outer scales have the richest quercetin concentration, minimal onion peeling is recommended during food preparation [199]. In apples, quercetin glycosides are the highest early in the season and decrease to a steady level during maturation [200]. The fruits exposed to the sun have greater levels of quercetin glycosides than shaded fruits [201].

The absorption unit of quercetin is the “aglycone”. Before absorption into the enterocyte, bound chemical groups, such as sugars, must be removed. This is accomplished by brush-edge enzymes, such as lactase floridin hydrolase (LPH), which removes glucose groups from flavonols [202]. A safe quercetin dose is 945 mg/kg. A toxic dose of quercetin can cause emesis, hypertension, nephrotoxicity, and a reduction in serum potassium. The distribution and elimination half-life of intravenous quercetin is 0.7–7.8 min and 3.8–86 min, respectively. Its clearance is 0.23–0.84 L/min/m^2^, and the volume of distribution is 3.7 L/m^2^ [203]. Several studies have reported that quercetin has antioxidant and anti-inflammatory effects, as well as beneficial applications in coronary and pulmonary artery vasodilatation; it also reduces blood pressure [204,205,206]. Although it has been suggested that gender affects quercetin bioavailability, there is no clear evidence of this effect [207].

On the other hand, studies have reported the effect of quercetin in two models of PAH. In an MCT-induced PAH model (60 mg/kg intraperitoneally), quercetin was administered by gastric gavage at a dose of 10 mg/kg/d for 11 days after 21 days of MCT administration. The results showed that quercetin decreased the PAP, RV hypertrophy, vascular remodeling, and mortality of the rats (Figure 2). Further, quercetin exerted effective vasodilator effects on isolated pulmonary arteries and inhibited cell proliferation and apoptosis. These effects are possibly associated with decreased expression of the 5-HT2A receptor and Akt and S6 phosphorylation and a partial restoration of voltage-activated potassium channel currents [208]. In another study, PAH was induced by chronic hypoxia in a hypobaric chamber for four weeks (10% O_2_ and 90% N_2_). Like MCT-induced PAH, quercetin decreased RV hypertrophy, inhibited PASMC proliferation, and increased the apoptosis of PASMC. In addition, quercetin may be associated with increasing the cyclin D1 protein levels and decreasing the protein expression of cyclin B1 and Cdc2 and reducing the expression of MMP2, MMP9, and CXCR4 integrin β1 and integrin α5 expression. Moreover, quercetin inhibited the activation of the TrkA/AKT signaling (Figure 3) cascade, which resulted in decreased PASMC migration and the induction of apoptosis [209].

### 4.12. Beet Juice

Beet juice is commonly used as a supplement because of its high quantity of inorganic nitrate, a compound found naturally in vegetables and processed meats [210]. Nitrate is believed to produce cardiovascular benefits, such as protecting against myocardial ischemia/reperfusion, vascular dysfunction, and hypertension in mice and rats [211,212]. A study found decreased nitrate and increased nitrite contents in home-made raw beet, carrot, and radish juices after 48 h of storage at ambient temperatures. However, the nitrite contents of beet and the other raw juices remained high after 24 h of storage at ambient temperatures. Therefore, the consumption of home-made vegetable-based raw juices is recommended immediately after their preparation [213]. On the other hand, cooking methods can affect the properties of beet. Studies determined that beet boiling offers the highest value of the possible cooking methods, while steam is the most suitable method for the retention of the phytochemicals and antioxidant capacity in beet [214].

When beet juice is ingested, nitrates accumulate in the saliva and come into contact with symbiotic bacteria on the dorsal surface of the tongue, where inorganic nitrate is reduced to nitrite through bacterial nitrate reductases (xanthine oxidase). This saliva, rich in nitrogen compounds, reaches the stomach where a small part of the nitrite is reduced to NO through a nonenzymatic reaction, which is favored by the acidic environment of the stomach [215]. However, the greatest portion of nitrate and nitrite is rapidly absorbed by the stomach and duodenum where it subsequently enters systemic circulation. Between 20% and 25% of nitrate is reabsorbed from the bloodstream and concentrated in the salivary glands to later function as a substrate for the bacteria mentioned above, which produce nitrite [216]. This mechanism increases the concentration of these ions in the plasma (up to 182 ± 55 µM after 1–2 h for nitrate and 373 ± 211 nM after 2–3 h for nitrite), which facilitates the production of NO in the wall of blood vessels and erythrocytes. A process that is carried out by means of enzymatic reduction mechanisms, such as xanthine oxidoreductase, enzymes of the respiratory chain, and aldehyde oxidase, as well as those of a nonenzymatic nature, such as hemoglobin, deoxygenated myoglobin, protons, vitamin C, and polyphenols. This reduction process is stimulated under conditions of low oxygen availability and an acidic pH, which allows NO synthesis to be localized at certain specific times. This increase in NO concentration promotes vasodilation through different cellular mechanisms, such as the cGMP/PKG pathway and hyperpolarization/relaxation after activation of the K^+^ channels, thereby causing a decrease in blood pressure that produces muscle relaxation in the endothelium. Finally, nitrate is excreted in the urine by the kidneys [217,218]. To the best of our knowledge, the effects of administering beet juice have not been studied between genders.

Evidence indicates that inorganic nitrite and nitrate can improve vascular remodeling, pulmonary circulation, and RV hypertrophy in PAH models (Figure 2) [219,220,221,222,223].

Currently, beet juice supplementation promotes beneficial effects against PAH, such as RV hypertrophy, pulmonary arterial remodeling, and RVSP [224]. In this experiment, PAH was induced by MCT (60 mg/kg subcutaneously), and all the measurements were made over four weeks. Beet juice was prepared by dissolving beet juice powder at a concentration of 1 g/L or 10 g/L in drinking water. This supplementation started from the day of MCT injection, one week before MCT injection, and two weeks after MCT injection. The authors found that beet juice administered at low doses exerted protective effects against RV hypertrophy, pulmonary arterial remodeling, and RVSP. However, a low dose was ineffective against functional and morphological alterations in circulation. Beneficial effects were not observed at high doses. Finally, the habitual ingestion of beet juice could be a potential option for preventing PAH, but not in a critical status of the pathology.

### 4.13. Chlorogenic Acid

Chlorogenic acid ((1S,3R,4R,5R)-3-{[(2E)-3-(3,4-dihydroxyphenyl)prop-2-enoyl]oxy}-1,4,5-trihydroxycyclohexanecarboxylic acid) is widely distributed in plants and is commonly found in traditional Chinese medicine [225]. Chlorogenic acid is a phenolic compound from the hydroxycinnamic acid family and is commonly available in human food in the form of drinks such as coffee and tea. Several polyphenols were isolated in coffee, especially chlorogenic acid, which is considered to be a well-known antioxidant agent [226]. It is one of the main polyphenols in the human diet and can be found in foods and herbs such as apple, artichoke, betel, burdock, carrot, coffee bean, eggplant, Eucommia, grape, honeysuckle, kiwi fruit, pear, plum, potato, tea, tomato, and wormwood [227,228,229,230,231,232,233]. Chlorogenic acid content increases with a longer cooking time. However, a high temperature over 250 °C causes a decrease in this compound. To ensure high levels of chlorogenic acid in eggplant, grilling, roasting, and baking are recommended. Stored chlorogenic acid decreases within four weeks in fresh eggplant, but an increase of chlorogenic acid in heat-treated eggplant was observed within the same period [234]. Roasted coffee contains a similar proportion of chlorogenic acid compared to unroasted coffee. There is, moreover, no difference in the chlorogenic acid level between instant coffee and coffee made with caffeine. Furthermore, decaffeination did not have effect on the chlorogenic acid content. Finally, coffee selection may have a profound influence on an individual’s intake of chlorogenic acid [235]. Studies reported that an intake of 400 mg chlorogenic acid can reduce systolic and diastolic blood pressure in humans [236].

Chlorogenic acid is absorbed and hydrolyzed in the small intestine where it inhibits Na+-dependent glucose absorption and enhances the release of gastric inhibitory polypeptide. Like other phenolic acids, chlorogenic acid is poorly excreted in the bile and intestinal lumen and has the potential to alter the balance of the gut microbiota [237]. The bioavailability of this acid appears to be dependent on its metabolism in the gut microflora [238]. In addition, to the best of our knowledge, studies that compare chlorogenic acid administration between genders are lacking.

Affective activities against hypertension in cardiovascular diseases have been observed in individuals who received coffee bean extract for 28 days, which reduced the blood pressure without any adverse effects [239].

A recent study determined the possible mechanisms involving chlorogenic acid on hypoxia-induced PASMC proliferation via the pathway c-Src [240]. c-Src is a cellular nonreceptor tyrosine kinase that is a proto-oncogene, whose exposure to stimuli such as hypoxia results in a conformational change and subsequent activation. Subsequently, c-Src activation increased VSMC proliferation [241]. For this study, PASMCs were obtained from the pulmonary arteries of rats and placed in a modular chamber (hypoxia 3% O_2_). Additionally, PAH was induced via the intraperitoneal injection of MCT (50 mg/kg). Rats were treated with 50, 100, and 200 mg/kg of cholorogenic acid by oral gavage once daily for 28 days. The results showed that cholorogenic acid reduced hypoxia-induced hypoxia-inducible factor 1α (HIF-1α) expression. Further, cholorogenic acid effectively attenuated c-Src and the physical coassociation of c-Src/Shc/Grb2 and ERK2 phosphorylation in PASMCs (Figure 3). Finally, in rats, a high dose (200 mg/kg per day) of cholorogenic acid inhibited the pulmonary arterial hyperplasia of MCT-induced PAH.

To use cholorogenic acid for the treatment of PAH, new experimental evidence is necessary to support the postulated beneficial properties outlined above. For this reason, it would be convenient to start by reviewing the in vivo effects of RV hypertrophy and subsequently the hemodynamic parameters. However, the inclusion of products high in cholorogenic acid into a habitual diet may protect from cardiovascular diseases, such as PAH.

### 4.14. Fatty Acids

In the body, fatty acids can be present as free fatty acids that link to glycerol to form triacylglycerol, diacylglycerol, or monoacylglycerol or as a composition of membrane phospholipids [242]. Polyunsaturated fatty acids include at least two double carbon-to-carbon bonds in their long hydrocarbon chains. This chemical characteristic makes the fats pack less tightly, so they tend to be liquid at room temperature, rather than solid like many saturated fats. Polyunsaturated fatty acids can be omega-3, where the first double bond is three carbons away from the methyl-carbon end of the molecule, or omega-6. Fish and plant oils are often rich in polyunsaturated fatty acids. Fish is rich in omega-3, and plant oils are rich in omega-6. Alpha-linolenic acid (omega-3) and linoleic acid (omega-6) are two polyunsaturated fatty acids that are essential nutrients in humans [243].

Several studies have shown that cooking methods change the number of fatty acids. Recently, i different cooking methods were evaluated on fillets (deep drying, grilling, baking in foil, and steaming). The results demonstrated that steaming and baking in foil are the best cooking methods for the retention of DHA (docosahexaenoic acid, 22:6n-3) and EPA (eicosapentaenoic acid, 20:5n-3) in fillets [244]. In addition, for foal meat, the best cooking techniques are grilling and roasting because they do not affect the amino acid concentration from a nutritional standpoint [245]. The data suggest that gender affects fatty acid metabolism. Several studies have found that sexual hormones can modify the levels of polyunsaturated fatty acids (omega-3) contained in plasma and tissue, causing lower fatty acid levels in females; however, sex explains only 2% of the variability of fatty acids in plasma [246,247].

The most common dietary ingredient used to enrich poultry products such as meat and egg with omega-3 fatty acids is flaxseed, which is a rich source of omega-3 (>50% α-linolenic acid). α-linolenic acid is an essential fatty acid for humans because it cannot be synthesized from saturated fatty acids (omega-9 monounsaturated fatty acids and omega-6 polyunsaturated fatty acids) [248]. The beneficial effects of polyunsaturated fatty acids have been shown in the prevention and treatment of anti-inflammatory disorders, coronary arterial disease, and hypertension [249].

In a study on PAH, the oxygen demanded and provided by the heart and lungs led to a dramatic increase in cardiovascular disorders. Broilers produced at high altitudes induced significant hypobaric hypoxia, which led to the development of PAH [250]. Recently, the effects of flax oil and soy oil (used as sources of omega-3 and omega-6) were evaluated in broiler chickens reared at 2100 m above sea level [251]. These oils were included at 50 mg/kg in the basal diet. In this protocol, only flax oil presented higher serum concentrations of NO. The mortality from PAH, RV hypertrophy, and hepatic lipogenesis decreased in birds that received flax oil. Finally, the supplementation of omega-3 fatty acids in the basal diet could increase the circulatory levels of NO, thereby suppressing hepatic lipogenesis and reducing the mortality of PAH.

Docosahexaenoic acid is an omega-3 fatty acid that is present in high concentrations in salmon, herring, and trout. Some studies have reported that docosahexaenoic acid inhibits the development of inflammation and cardiovascular events [252]. Further, in rat coronary arteries, the smooth muscle cells are a potent activator of calcium-dependent K^+^ channel currents, which promote the dilation of isolated small coronary arteries [253].

In another study, tissue samples of patients with PAH were analyzed to evaluate the depolarized effects of docosahexaenoic acid on PASMCs via the activation of the calcium-dependent K^+^ channel [254]. The results showed that docosahexaenoic acid caused a dose-dependent activation of calcium-activated K^+^ current in human PASMCs and an endothelial-dependent relaxation of the pulmonary arteries. On the other hand, docosahexaenoic acid ameliorated the RVSP in chronic hypoxia-induced PAH in mice. Ultimately, these findings suggest that docosahexaenoic acid activates the PASMC calcium-dependent K^+^ channel, leading to vasorelaxation in PAH.

These recent investigations demonstrate that polyunsaturated fatty acids have a protective effect in cardiovascular diseases like PAH (Figure 2). Determining supplementation with these fatty acids without causing adverse effects in humans is key to improving quality of life. The introduction of food rich in polyunsaturated fatty acids into the diets of patients diagnosed with PAH could possibly contribute to improvement of the condition and enhance the effects of the drugs. However, more research is required to determine the possible signaling pathways in which they participate and, in the future, to promote supplementation as a treatment for PAH.

### 4.15. Sulforaphane

Sulforaphane (1-Isothiocyanato-4-(methanesulfinyl)butane) is an active molecule found particularly in broccoli and cauliflower extracts that acts as an inducer of NAD(P)H quinone oxidoreductase 1 and quinone reductase activity [255]. The flavonoid content in vegetables such as broccoli can vary depending on the cooking method. In a study, raw broccoli was cooked by boiling, steaming, and microwaving. In general, the results showed that boiling yielded a significant loss of all flavonoids, while steaming and microwaving led to minor losses or even increases in the flavonoids [256]. Specifically, the contents of sulforaphane and sulforaphane nitrile production were evaluated in the broccoli. Irrespective of time, steaming resulted in a significantly greater retention of sulforaphane while boiling, and microwave cooking resulted in significant losses of sulforaphane. Potentially beneficial compounds and optimal sulforaphane ingestion can be obtained by eating raw or lightly steamed broccoli [257].

Sulforaphane is formed through the hydrolyzation of its glucoraphanin precursor by myrosinase when plant tissue is damaged. Intestinal bacteria can, moreover, form small amounts of sulforaphane when the myrosinase plant is inactive [258]. Differences in the sulforaphane bioavailability among ingested forms of broccoli have largely been attributed to differences in myrosinase activity. A previous study found that subjects that consumed raw broccoli or broccoli sprouts with intact myrosinase had a high percentage of sulforaphane and its metabolites (32–80%), and those that consumed foods like cooked cruciferous vegetables with inactive myrosinase had a low percentage of sulforaphane (10–12%) [259,260]. Moreover, after the ingestion of broccoli sprout, sulforaphane was absorbed, reaching its peak in the plasma at 1 h [261]. Sulforaphane is conjugated with glutathione and cysteinyl glycine and also with other thiols in the liver, which can be transported through membrane transporters like P-glycoprotein [262]. Sulforaphane is slowly eliminated with a half-life of 40 h and remains detectable in plasma 24 h after the intake of broccoli [263]. Sulforaphane-glutathione, sulforaphane-cysteine glycine, sulforaphane-cysteine, and sulforaphane N-acetylcysteine are excreted in the urine [264]. A lack of information on the differences in sulforaphane activity between males and females is evident; thus, more clinical trials that compare these variables are needed.

Sulforaphane is an isothiocyanate and potent Nrf2 activator, which reacts with the thiol groups of Keap1 and promotes Nrf2 dissociation from Keap1 [265]. Sulforaphane-induced Nrf2 is involved with the anti-inflammatory pathways, protective effects on the light ventricle in diabetic cardiomyopathy, angiotensin II-induced cardiomyopathy, and antifibrotic effects in the lungs [266,267,268,269].

The protective effects of these properties have been reported to occur in RV hypertrophy in PAH via the Nrf2 signaling pathway [270]. In this work, PAH was induced in male C57BL/6J mice by weekly subcutaneous injections of a VEGFR inhibitor (SU5416, 20 mg/kg), accompanied by four weeks of hypoxia (10% oxygen). Sulforaphane was given subcutaneously daily at 0.5 mg/kg for 5 days per week for four weeks. The results showed that sulforaphane prevented RV dysfunction and remodeling and reduced RV inflammation and fibrosis (Figure 2). Further, it upregulated Nrf2 expression and its downstream gene NQO1 (Figure 3). Interestingly, this administration with sulforaphane reduced the inflammatory mediators leucine-rich repeat and pyrin domain-containing 3. Finally, these findings show that sulforaphane attenuates or prevents RV and lung injury in a murine model via the Nrf2 signal pathway and may be candidate target for strategies to prevent or reverse PAH.

### 4.16. Allicin

Allicin (S-Prop-2-en-1-yl prop-2-ene-1-sulfinothioate) is a natural sulfur-containing compound that is responsible for the typical smell and taste of freshly cut or crushed garlic (Allium sativum L.) [271]. The volatile bioactive compounds of garlic, such as allicin, can be modified using techniques that involve heat transfer. Initially, studies showed that 60 s of microwave heating or 45 min of oven heating can block garlic’s anticarcinogenic abilities. In contrast, this study suggested allowing crushed garlic to stand for 10 min before being treated with microwave heating for 60 s to prevent the total loss of anticarcinogenic activity [272]. Recently, studies showed that, after crushing garlic, the number of organic sulfides increased over time, but the amount of allicin was not influenced. Increasing the heating temperature enhanced the formation of diallyl sulfide (DAMS) and methyl propenyl disulfide (MPDS) but had no effect on the formation of allicin. In addition, with the exception of DAMS, ten minutes of boiling hampered the formation of sulfur compounds when the heat-treated garlic was stored at room temperature. Consequently, it is advisable to consume the prepared meal as soon as possible because the further formation of organic sulfides is limited [273]. Allicin is an unstable compound that rapidly decomposes into other oil and water-soluble organosulfur compounds. Absorption depends on many factors, such as physicochemical properties like molecular size, steric configuration, solubility, and pKa [274]. Clinical trials comparing allicin between genders are needed to demonstrate any significant differences in its function. When garlic cloves are crushed or macerated, allicin is naturally produced from the stable precursor S-allyl cysteine-S-oxide (alliin) via the action of the enzyme alliinase [275].

Several studies have reported the protective actions of allicin against coronary artery disease and hypertension [275,276,277]. Interestingly, the blood-pressure-lowering properties of garlic come from the bulb portion, specifically sulfur-containing compounds like allicin.

In another study on PAH, rats received a single subcutaneous injection of 50 mg/kg of MCT 3 days after feeding on fermented garlic extract. The fermented garlic extract attenuated the systolic pressure and atrial natriuretic peptide concentration in the RV, as well as the endothelial damage and medial hypertrophy of the pulmonary arterioles and pulmonary fibrosis. This extract had a protective effect by inducing an anti-inflammatory response via a decrease in inflammatory proteins such as VCAM-1 and MMP-9. These two molecules are produced by inflammatory cytokines such as TNF-α and IL-1β and bind monocytes to damaged arteries, where the recruitment of these monocytes can facilitate costimulation and the transmigration of inflammatory cells to the extracellular matrix, as well as activate uncontrolled angiogenesis [278,279]. Finally, allicin increased the expression of PKG and eNOS in lung tissue [280]. This work proposed the signaling pathway NO-sGC-PKG.

## 5. Future Directions

Studies that evaluate the effects of these nutraceuticals in patients with PAH and their intersections with current therapies.Determining the dose that exerts beneficial effects in animal models to calculate the amount by which people diagnosed with PAH should obtain from these nutraceuticals.Test the use of these nutraceuticals as part of an alternative diet for PAH patients.Study the involvement of the signaling pathways responsible for the development of PAH, such as BMPR-2 (Bone morphogenic protein receptor type 2) and ALK-1 (Activin A receptor type II-like kinase-1), in response to nutraceutics.

## 6. Conclusions

In conclusion, nutraceuticals exert beneficial effects on PAH through various signaling pathways that are similar to those of classic drugs used for the management of this pathology. In addition, these nutraceuticals could have the ability to enhance the effects of these drugs and thus prolong the life expectancy of these patients, in addition to acquiring new eating habits with the introduction of these nutrients in the diet. Finally, it is important to highlight that a potential therapy could combine nutraceuticals and drugs to improve quality of life.

## Figures and Tables

**Figure 1 ijms-21-04827-f001:**
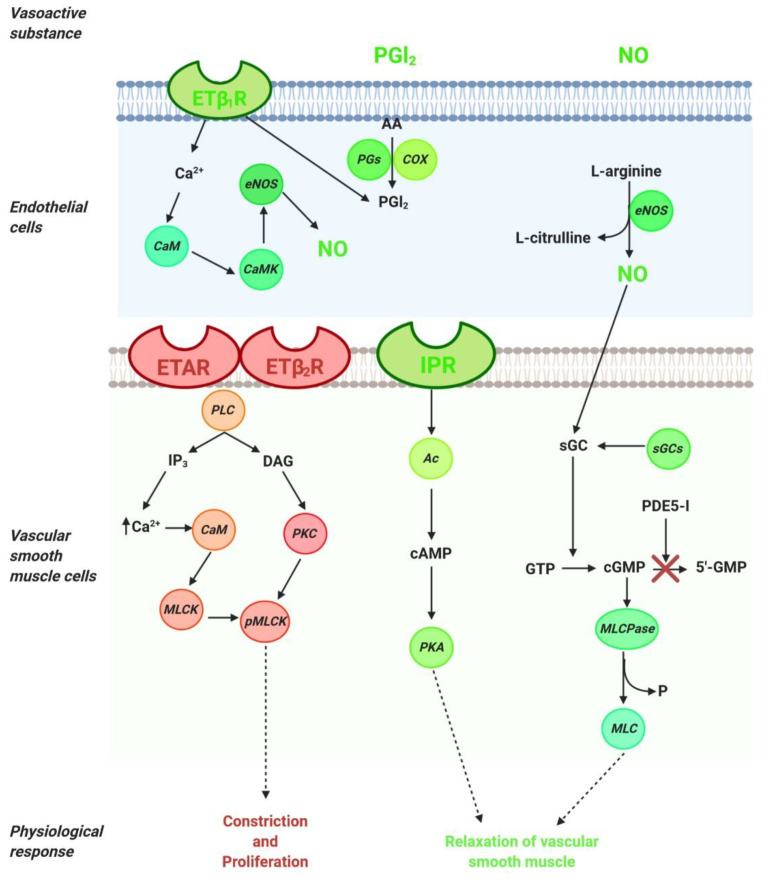
Signaling pathways in the regulation of pulmonary vascular tone. Arachidonic acid (AA); adenylyl cyclase (Ac); calmodulin (CaM); calmodulin kinase (CaMK);cyclooxygenase (COX); cyclic adenosine monophosphate (cAMP); cyclic guanosine monophosphate (cGMP); diacylglycerol (DAG); endothelial nitric oxide synthase (eNOS); endothelin 1 (ET-1); endothelin type A receptor (ETAR); endothelin type B1 receptor (ETB1R); endothelin type B2 receptor (ETB2R); guanosine monophosphate (GMP); inositol trisphosphate (IP3); I-prostanoid receptor (IPR); myosin light chain (MLC); myosin light chain kinase (MLCK); myosin light chain phosphatase (MLCPase); nitric oxide (NO); protein kinase A (PKA); prostaglandin synthase (PGs); protein kinase C (PKC); phosphodiesterase (PDE)-5 inhibitors (PDE-5i); phospholipase C (PLC); phosphorylated myosin light chain (pMLC); soluble guanylate cyclase stimulator (sGCs).

**Figure 2 ijms-21-04827-f002:**
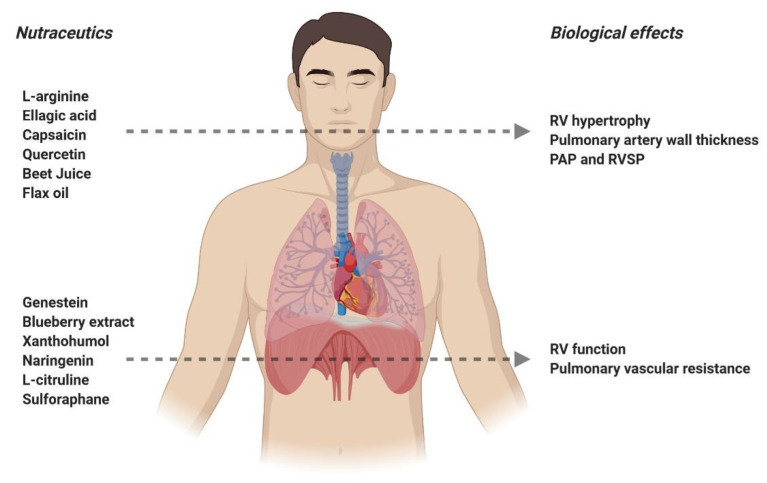
Biological effects of nutraceuticals. L-arginine, ellagic acid, capsaicin, quercetin, beet juice, and flax oil have a protective effect against RV hypertrophy and result in greater pulmonary arterial wall thickness and an increase in pulmonary arterial pressure (PAP) and RVSP. Figure image created with Biorender.com.

**Figure 3 ijms-21-04827-f003:**
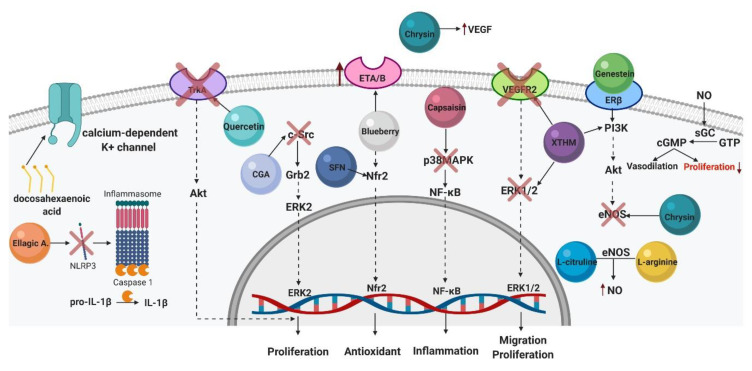
Participation of nutraceuticals in the signaling pathways involved in the development of PAH. TrkA: Tropomyosin receptor kinase A; Akt: Protein Kinase B; c-Src: Proto-oncogene tyrosine-tyrosine-protein kinase Src; Grb2: Growth factor-bound protein 2; ERK2: Extracellular-regulated kinase 2; Nrf2: Nuclear factor erythroid 2-related factor 2; ETA/B: Endothelin receptor A/B; VEGF: Vascular endothelial growth factor; p38MAPK: P38 mitogen-activated protein kinases; NFkB: Nuclear factor kappa B; VEGFR2: Vascular endothelial growth factor receptor 2; XTHM: Xanthohumol; ERK 1/2: Mitogen-activated protein (MAP) kinase; ERβ: Estrogen Receptor β; PI3K: Phosphatidylinositol 3-kinase; eNOS: Endothelial nitric oxide synthase; NO: Nitric oxide; sGC: Soluble guanylyl cyclase; cGMP: Cyclic guanosine monophosphate; and GTP: Guanosine triphosphate; Created with Biorender.com.

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
