# Peer review of "Nutraceuticals in the Treatment of Pulmonary Arterial Hypertension"

_ijms, 2020, doi:10.3390/ijms21144827_

Round 1

Reviewer 1 Report

  1. The incidence of PAH is different between male and female and survival is paradoxical worse in male. Is there any differences about nutraceuticals between gender?
  2. ETBR acts to clear ET-1 and mediate the endothelial cells vasodilation through the production of PGI2 and NO, but induces vasoconstrictive and proliferative responses in VSMCs. Author should explain the effect of ETBR on PAH.
  3. Since ERAs inhibit both ETAR and ETBR, which reduced the production of PGI2 and NO associated with the effect of ETBR on endothelial, and increase the inhibition of vasoconstriction in VSMC. Author should make further descripiton about the mechanism of ERAs on PAH therapy. Is it a composite result between vasodilation and vasoconstriction?
  4. The evidence of genistein to improved PAH on the MCT-induced PAH animal model comes from a daily subcutaneous dose of 1 mg/kg genistein for 10 145 days. Is the effect from parenteral route equals to enteral dietary management? Is there any report supporting the advantage of genistein from enteral route?
  5. Several participations of nutraceuticals in signaling pathways are associated with eNOs and the production of NO. However, they are substrates rather than direct agonist of NO production compared to ERAs. Is there any study to compare the improvement between dosage titration and add-on nutraceuticals?
  6. Most of nutraceuticals are existed in foods, and the stabilization of these materials in different temperature, states of matter, and cooking method have inadequate information in this review. Is there any report about this issue?
  7. These nutraceuticals are extracted form plants or foods, but the actual practice in daily life is to take cooked plants and foods. Is there any study about the benefit of taking plants and foods with plenty of these nutraceuticals on the improvement of PAH?
  8. Many nutraceuticals are given through subcutaneous route, such as Sulforaphane, Allicin, Capsaicin, etc. However, there are other routes including enteral and inhalation or external application. Is there any study about the application of nutraceuticals in different routes?
  9. Is there any interaction between nutraceuticals, or between nutraceuticals and medications? Is there any contraindications or adverse effects of taking nutraceuticals ?
  10. Is there any study discussing how to quantitative nutraceuticals, such as, how much fruits, vegetables or meats should be intook per day?

Author Response

Response to reviewers.

Reviewer 1:

1.- The incidence of PAH is different between male and female and survival is paradoxical worse in male. Is there any differences about nutraceuticals between gender?

Answer: Genistein acts in both male and female cell, and for this reason, we added in the review this paragraph: In this context, genistein has exerts effects on breast and prostate cancer through direct competition with estrogen to bind to the estrogen receptor; prolongated exposure decreases these receptors’ expression, thus lowering endogenous estrogen responsiveness [8]. P. 7, L. 295-298.

In fatty acids section was including the following paragraph: The data suggest that gender affects fatty acid metabolism. Several studies have found that sexual hormones can modify the levels of polyunsaturated fatty acids (omega-3) contained in plasma and tissue, causing lower fatty acid levels in females; however, sex explains only 2% of the variability of fatty acids in plasma [246,247]. (P. 17, L. 3511-3515).

Although it has been suggested that gender affects quercetin bioavailability, there is no clear evidence of this effect [207]. (P. 15, L. 2872-2873).  

However, other nutraceuticals included in this review found no studies comparing their administration between genders.

We considered including additional information on the incidence and survival of PAH between men and women.

Although the incidence of PAH is higher among women, there are no nutraceutical treatment studies that compare the efficacy of such therapies between both genders. The degree of damage caused by PAH in both genders is also different; vascular remodeling and RV hypertrophy, for example, are more evident in male than female rats [1,2]. Studies suggest that these differences are related to sex hormone levels, where estrogens exert protective effects against this pathology [54–56]. Previous studies have demonstrated that estrogen plays an important role in regulating vascular oxidative stress in MCT-induced PAH [57]. On the other hand, in an angioproliferative model of severe PAH female rats, this pathology was associated with angioproliferative changes in the pulmonary arteries but with absence of inflammatory and fibrinogenic processes in both the lung and heart. This was correlated with preserved RV function and may be an important factor for the better survival rates in women [58]. Based on these studies, nutraceutical treatments may offer a greater protective effect in male rats than female rats due to the different degree of damage caused by PAH between genders. (P. 5, L. 550-562).

2.- ETBR acts to clear ET-1 and mediate the endothelial cells vasodilation through the production of PGI2 and NO, but induces vasoconstrictive and proliferative responses in VSMCs. Author should explain the effect of ETBR on PAH.

Answer: ETAR receptor is located predominantly on VSMCs and mediates vasoconstriction, as well as proliferation, hypertrophy, cell migration, and fibrosis. In contrast, ETBRs are found on both endothelial and VSMCs. ETB2R activation on smooth muscle cells produces vasoconstriction; however, ETB1R activation on endothelial cells leads to vasodilation and antiproliferation by increasing NO and prostacyclin production.  

Theory suggests that blocking ETARs may be a better therapeutic strategy. However, in clinical practice, blocking of selective and non-selective ETRs showed beneficial effects on hemodynamics, RV  hypertrophy, remodeling of the pulmonary artery, the endothelial function of the pulmonary vessels, and increases survival.

As PAH progresses, the expression of ETRs increases in smooth muscle cells, that leads to vasoconstriction and proliferative effects, therefore, there is a possibility that is blocking of these receptors, mainly in muscle, favors the activation and interaction of ET-1 with its endothelial ETB1R to induce vasodilator and antiproliferative effects.

On the other hand, as the reviewer correctly points out in commentary three, it cannot be ruled out that the beneficial effects may be due to a balance between the effects mediated by the blocking of ETAR and ETBR in VSMCs.

3.- Since ERAs inhibit both ETAR and ETBR, which reduced the production of PGI2 and NO associated with the effect of ETBR on endothelial, and increase the inhibition of vasoconstriction in VSMC. Author should make further description about the mechanism of ERAs on PAH therapy. Is it a composite result between vasodilation and vasoconstriction?

Answer: We have included additional information and figure (Figure 1) about the mechanisms throughout ET-1 induces its physiological effects. ETAR receptor is located predominantly on VSMCs and mediates vasoconstriction, as well as proliferation, hypertrophy, cell migration, and fibrosis. In contrast, ETBRs are found on both endothelial and VSMCs. The ETB2R activation on smooth muscle cells produces vasoconstriction; however, ETB1R activation on endothelial cells leads to vasodilation and antiproliferation by increasing NO and prostacyclin production. As PAH progresses, the expression of ETRs increases in smooth muscle cells, that leads to vasoconstriction and proliferative effects, therefore, there is a possibility that is blocking of these receptors, mainly in muscle, favors the activation and interaction of ET-1 with its endothelial ETB1R to induce vasodilator and antiproliferative effects. Furthermore, it cannot be ruled out that the beneficial effects observed with the use of ETR blockers may be due to a balance between vasoconstriction and vasodilation.

4.- The evidence of genistein to improved PAH on the MCT-induced PAH animal model comes from a daily subcutaneous dose of 1 mg/kg genistein for 10 days. Is the effect from parenteral route equals to enteral dietary management? Is there any report supporting the advantage of genistein from enteral route?

Answer: Mostly, all reports of genistein administration to treat PAH are by a direct dose. Total isoflavone, daidzein, and genistein contents are highest in protein ingredients that have not been subjected to heat treatment or followed fermentation steps. The simplest soy food product is the soy milk (basically the water extract of soybean) and has considerably higher total isoflavone than cooked. This information is from the study mentioned below, and there is no evidence to be used in PAH.

These sentences were added: There is evidence that 50 mg/day genistein administered orally offers the best clinical efficacy [85]. Optimum steady-state serum isoflavone concentrations consumed throughout the day are more effective than those from a single highly enriched product [86]. To find another way to supply genistein, Rassu et al. (2008) used an intranasal method to deliver genistein. Genistein-loaded chitosan nanoparticles were successfully obtained and were able to promote the passage of genistein through the nasal mucosa. These results are a promising as a way to supply genistein to the blood flow, but further investigations need to be performed. (P. 7, L. 1035-1041)

5.- Several participations of nutraceuticals in signaling pathways are associated with eNOs and the production of NO. However, they are substrates rather than direct agonist of NO production compared to ERAs. Is there any study to compare the improvement between dosage titration and add-on nutraceuticals?

Answer: NO production is due to an enzyme, not a receptor, as in endothelin, and enzymes are always activated through a precursor or specific substrate. In the case of the eNOS, its induction is by electron transfer. How could it be increased or regulated the electron transfer in a dose-response manner to activate the enzyme? Although NO has extremely short half-life, limited water solubility, and radical nature, so its induction is not easy to induce and maintain, and a constant production can be prooxidant. However, it has been described that calcium can be an activator of eNOS, but it is also a very important and promiscuous element involved in many other cellular functions.

On the other hand, berberine is a good inductor of eNOS via AKT. We found several studies where they used berberine to revert some parameters of PAH in animal models and endothelial cells.

For this reason, we added a new section, including all the information about berberine and PAH. (P. 8, L. 1295-1469).  

6.-Most of nutraceuticals are existed in foods, and the stabilization of these materials in different temperature, states of matter, and cooking method have inadequate information in this review. Is there any report about this issue?

Answer: Cooking methods induce significant changes in chemical composition, alternating the concentration and bioavailability of bioactive compounds in these nutraceuticals. However, studies have been reported positive effects depending upon differences in process conditions, as well as morphological and nutritional characteristics of these foods. We included this information in each nutraceutical.

7.-These nutraceuticals are extracted form plants or foods, but the actual practice in daily life is to take cooked plants and foods. Is there any study about the benefit of taking plants and foods with plenty of these nutraceuticals on the improvement of PAH?

Answer: There are currently no studies investigating the consumption of cooked plants and foods to improve the symptoms caused by PAH. Despite having beneficial effects in animal models, the evidence in patients is limited. The ESC / ERS Guidelines for the diagnosis and treatment of PAH do not contain nutritional information. On the other hand, the Dietary Approaches to Stop Hypertension (DASH) is a specific dietary intervention useful for the treatment of hypertension [59]. This diet recommends the consumption of carbohydrates from green leafy vegetables (broccoli and spinach), whole grains (wheat, millets and oats), low glycemic index fruits, legumes, and beans, as well as fats (olive oil, avocados, nuts, and fish) and animal protein from lean meats, low-fat dairy, eggs, and fish. Compared to PAH, this field of therapeutic intervention is not fully studied (P. 5, L. 563-571).

8.-Many nutraceuticals are given through subcutaneous route, such as Sulforaphane, Allicin, Capsaicin, etc. However, there are other routes including enteral and inhalation or external application. Is there any study about the application of nutraceuticals in different routes?

Answer: The route of administration of these nutraceuticals is not discussed in this study. However, in several other studies, the administration of nutraceuticals was carried out by oral administration (through drinking water, basal diet, or tube) and injection (subcutaneous and intraperitoneal). In PAH models, no reports were found for other routes of nutraceutical administration, such as an enteral route or inhalation. In this field, there are no studies investigating the application of nutraceuticals in different routes. However, as mentioned above, nutraceuticals are commonly purchased from food and consumed via the oral route. On the other hand, routes of administration have various advantages and disadvantages. Injection ensures a controlled dose of locally administered nutraceuticals. However, in rats, there is a limited volume that can be injected. On the other hand, in patients with PAH, the oral consumption of nutraceuticals can be controlled autonomously. In PAH models, oral administration by gastric tube allows a more rigorous control of the administered dose compared to the nutraceuticals present in the basal diet [60]. However, oral administration must take into account the stability of the compound to preserve its bioactivity [61]. (P. 6, L. 803-815).

9.-Is there any interaction between nutraceuticals, or between nutraceuticals and medications? Is there any contraindications or adverse effects of taking nutraceuticals?

Answer: Some foods may alter the absorption, distribution, biotransformation, and excretion of drugs. Moreover, various interactions may have an influence on the success of drug treatment. That is to say, interactions are not always harmful to therapy and in some cases can be used to improve the drug’s effects [62]. Many medicinal plants and natural compounds have beneficial effects in the treatment of PAH [63]. Interestingly, these compounds do not have more side effects than chemical drugs [64]. In humans, the oral administration of quercetin did not show any adverse effects at doses up to 730 mg per day over one month [65]. Specifically, nutraceuticals and other natural therapies for PAH have some interest as a way to enhance the effects of typical drugs used in PAH to improve the quality of life of patients suffering from this pathology. So far, there are no reports on the interactions between nutraceuticals and the drugs used to treat PAH. No adverse effects from the consumption of these nutraceuticals have been reported either. (P. 6, L. 816-826).

10.-Is there any study discussing how to quantitative nutraceuticals, such as, how much fruits, vegetables or meats should be in took per day?

Answer: Despite the various investigations including a known quantity of nutraceuticals in food [66], the recommended daily doses for patients with PAH have not been studied. For this reason, there is no study that analyzes the adequate amount of fruits, vegetables, cereals, and meats that a patient with PAH should consume daily. This field of research is still unknown. Some studies have tried to determine this relationship in other pathologies, such as in the habitual intake of polyphenols and the incidence of cardiovascular events. In this case, the intake of foods with a high content of polyphenols was positively associated with a decrease in cardiovascular risk. However, more clinical trials are needed to confirm this effect to establish precise dietary recommendations for this pathology [67]. (P. 6, L. 827-835).

Reviewer 2 Report

The paper seems with too much verbatim in this stage, for my taste. As being designated on IJMS, with focus on molecular structure, mechanism and functions- the present paper should be truly improved in the way expanding each sub-section of nutraceuticals with molecular structure, ligand-receptor mechanism and metabolic/biological activity/metabolomics response; kinetic data and mechanism may be also/equally considered; the citations should be accordingly enriched as such addition will be performed for each case of nutraceutics; all of these, when implemented, will truly contribute to enhance the reader and interested research overview – since this contribution is a Review – and will enhance the value of the present paper itself.

Author Response

Response to reviewers.

Reviewer 2:

1.- The paper seems with too much verbatim in this stage, for my taste. As being designated on IJMS, with focus on molecular structure, mechanism and functions- the present paper should be truly improved in the way expanding each sub-section of nutraceuticals with molecular structure, ligand-receptor mechanism and metabolic/biological activity/metabolomics response; kinetic data and mechanism may be also/equally considered; the citations should be accordingly enriched as such addition will be performed for each case of nutraceutics; all of these, when implemented, will truly contribute to enhance the reader and interested research overview – since this contribution is a Review – and will enhance the value of the present paper itself.

Answer: Thank you for the observations about the manuscript. In this version, we have extensively improved the manuscript according to the suggestions. The manuscript now has information for most of the included nutraceuticals about the ligand-receptor mechanism, metabolic and biological activity, as well as kinetic data. Furthermore, several new citations were added to the manuscript. We also included information about the nutraceutical stability, source, cooking methods, and the routes of administration.

Round 2

Reviewer 1 Report

no further comment

Reviewer 2 Report

Authors considerably improved on initial manuscript!